# An Anisotropic Damage-Plasticity Constitutive Model of Continuous Fiber-Reinforced Polymers

**DOI:** 10.3390/polym16030334

**Published:** 2024-01-25

**Authors:** Siyuan Chen, Liang Li

**Affiliations:** 1Wuhan National High Magnetic Field Center, Huazhong University of Science and Technology, Wuhan 430074, China; chen_siyuan@hust.edu.cn; 2State Key Laboratory of Advanced Electromagnetic Engineering and Technology, Huazhong University of Science and Technology, Wuhan 430074, China

**Keywords:** polymers, polymer composites, mathematical modeling, physical behavior, computational solid mechanics

## Abstract

Accurate structural analyses of continuous fiber-reinforced polymers (FRPs) are imperative for diverse engineering applications, demanding efficient material constitutive models. Nonetheless, the constitutive modeling of FRPs is complicated by the nonlinear behavior resulting from internal damages and the inherent plasticity. Consequently, this study presents an innovative anisotropic constitutive model for FRPs, designed to adeptly capture both the damage evolution and plasticity. All requisite parameters can be easily obtained through fundamental mechanical tests, rendering the model practical and user-friendly. The model utilizes the three-dimensional Puck criteria to determine damages, initiating the evolution process through a combination of continuum damage mechanics and linear stiffness attenuation methods. This evolution is coupled with a one-parameter plastic model. Subsequently, the numerical implementation method, integrated into ANSYS, is detailed. This emphasizes the Cauchy stress and consistent tangent stiffness solution strategy. Finally, the effectiveness of the developed model is demonstrated through comprehensive verification, encompassing existing biaxial tension and open-hole-tension tests conducted on carbon and glass FRP laminates. The simulation results exhibit a remarkable correspondence with the experimental data, validating the reliability and accuracy of the proposed model.

## 1. Introduction

Continuous fiber-reinforced polymers (FRPs) are progressively replacing metallic materials owing to their advantages in terms of their light weight, high stiffness, exceptional strength, and superior fatigue performance [1,2,3,4]. Realizing the full mechanical potential of FRPs necessitates precise structural analyses, which, in turn, require a robust constitutive modeling approach. However, the complexity of constitutive modeling for FRPs arises from distinctive load-dependent failure modes [5,6] and highly nonlinear deformation [7]. Concerning the former, numerous effective failure criteria for FRPs have been proposed [8], and predicting failure in FRPs poses minimal challenges in most engineering applications. The primary challenge in constitutive modeling lies in addressing the latter—the accurate representation of nonlinear deformation characteristics.

One contributing factor to the nonlinear behavior of FRPs is the presence of internal damage [9]. This damage initially manifests at the micro-level, occurring at a scale comparable to the fiber diameter. It involves phenomena such as fiber breakage, micro-cracking in the matrix, and localized fiber/matrix de-bonding [10]. With the progression of damage, these micro-level effects aggregate into macro-scale manifestations, encompassing matrix splitting, the wedge effect, and the scissoring effect [11]. This cumulative damage diminishes the load-carrying capacity of the lamina, ultimately leading to structural failure [12]. Collectively, both micro- and macro-damages contribute to the macroscopic nonlinear stress–strain response observed in FRPs.

As for micro-damages, the micromechanical modeling method serves as the most direct approach for describing its effects. While capable of providing a detailed micro-stress distribution, the computational expense associated with this method limits its practical application. Consequently, it is often employed specifically for capturing the elastic properties [9,13] and damage parameters [14,15] of FRPs. In contrast, the continuum damage mechanics (CDM) method offers greater computational efficiency. Numerous CDM models exist, among which the Cachan model stands out as one of the most widely utilized [16,17,18,19]. This model, utilizing the Gibbs potential, establishes a concise damage evolution based on energy-release rates [20,21]. As for macro-damages, a commonly adopted and convenient approach involves defining a stiffness degradation law. Depending on the specific degradation law applied, stiffness degradation methods can be categorized into linear [22,23,24], bilinear [25,26], and exponential degradation methods [27,28].

Another contributing factor of the nonlinear behavior is attributed to the irreversible deformation of the matrix [29,30]. Under matrix-dominated loading conditions or in the case of unidirectional laminates, damage has a minor effect on the nonlinear behavior [29]. In such cases, the tangent elastic moduli degrade rapidly at the very beginning (less than 10% for in-plane shear). Interestingly, the in situ modulus at the same stress level shows no apparent change during the unloading process [31,32,33]. This nonlinearity cannot be solely attributed to damages, as cracks are scarce and invisible [34,35]. These observations highlight the necessity for the constitutive model of FRPs to encompass plasticity.

Taking into account the aforementioned considerations, this research aims to propose an anisotropic constitutive model for FRPs, with the capacity to comprehensively capture micro-damage, macro-damage, and plasticity phenomena. This study elucidates the methodologies for damage determination and evolution and the characterization of plasticity. Additionally, detailed descriptions of the numerical implementation integrated into ANSYS and the model verification process are provided. In summary, the constructed model demonstrates satisfactory precision. All intrinsic parameters can be easily derived through fundamental mechanical tests, enhancing the model’s practicality and user-friendliness for composite analysis in engineering applications. To the best of the knowledge of the authors, there have been no reports on constructing the constitutive model of FRPs in this way to date.

## 2. Modeling Methodology

In light of the aforementioned characteristics of internal damage, the loading process of an FRP unidirectional lamina was delineated into three stages, as illustrated in Figure 1:Pure elastoplastic stage (OA segment): in this initial stage, the matrix remains devoid of any damage, and the nonlinearity is exclusively attributed to plasticity.Micro-damage stage (AB segment): As the loading progresses, micro-damages gradually emerge, introducing a subtle deviation in the elastic stress–strain response. The nonlinear behavior of FRPs contributes to both plasticity and internal micro-damages.Macro-damage stage (BC segment): Advancing to the final stage, macro-damages emerge, resulting in a noticeable deviation in the elastic stress–strain response. The nonlinear behavior of FRPs contributes to both plasticity and macro-damages.

Accordingly, the constitutive model should comprise three components: (1) Damage determination. This component evaluates the onset of micro-damages and their progression into macro-damages, specifically identifying points A and B. (2) Damage evolution. This component assesses the damage development in the AB and BC segments. (3) Plasticity evolution. This component assesses the evolution of plastic deformation in the entire loading process. Of course, each component should maintain an anisotropic form. Elaborated descriptions of each component are presented below.

### 2.1. Damage Determination

The three-dimensional (3D) Puck criteria [5,36], acknowledged as being among the most advanced failure criteria since the worldwide failure exercise (WWFE) [7,8,37,38], were employed to assess damage initiation. Rooted in Mohr’s fracture hypotheses, Puck’s criteria develop from the concept of intrinsically brittle materials. The criteria encompass both fiber fracture (FF) and inter fiber fracture (IFF). The latter specifically refers to a crack running parallel to the fibers through the layer thickness and is characterized by four distinct forms (Figure 2) [5,36].

The stress exposure for FF is [5]
(1)$$\left\{\begin{array}{l} f_{E,FF}=\frac{\sigma_{11}}{X_{11}^{t}},\;\mathrm{for}\;\sigma_{11}>0\\ f_{E,FF}=\frac{\sigma_{11}}{\left(-X_{11}^{c}\right)},\;\mathrm{for}\;\sigma_{11}<0 \end{array}\right.,$$
where ‘1’ denotes the fiber direction; $X_{11}^{t}$ and $X_{11}^{c}$ signify the longitudinal tensile and compression strengths, respectively.

The stress exposure for IFF is [5]
(2)$$f_{E,IFF}=\left\{\begin{array}{c} \sqrt{{\left[\left(\frac{1}{R_{⊥}^{t}}-\frac{p_{⊥\psi }^{t}}{R_{⊥\psi }^{A}}\right)\cdot \sigma_{n}\left(\theta \right)\right]}^{2}+{\left(\frac{\tau_{nt}\left(\theta \right)}{R_{⊥⊥}^{A}}\right)}^{2}+{\left(\frac{\tau_{n1}\left(\theta \right)}{R_{⊥∥}}\right)}^{2}}+\frac{p_{⊥\psi }^{t}}{R_{⊥\psi }^{A}}\sigma_{n}\left(\theta \right),\;\sigma_{n}\left(\theta \right)\geq 0\\ \sqrt{{\left(\frac{\tau_{nt}\left(\theta \right)}{R_{⊥⊥}^{A}}\right)}^{2}+{\left(\frac{\tau_{n1}\left(\theta \right)}{R_{⊥∥}}\right)}^{2}+{\left[\frac{p_{⊥\psi }^{c}}{R_{⊥\psi }^{A}}\sigma_{n}\left(\theta \right)\right]}^{2}}+\frac{p_{⊥\psi }^{c}}{R_{⊥\psi }^{A}}\sigma_{n}\left(\theta \right),\;\sigma_{n}\left(\theta \right)<0 \end{array}\right.$$
with
(3)$$\begin{array}{l} \sigma_{n}\left(\theta \right)=\sigma_{22}\cdot \cos^{2}\theta +\sigma_{33}\cdot \sin^{2}\theta +2\sigma_{23}\cdot \sin\theta \cos\theta \\ \tau_{nt}\left(\theta \right)=-\sigma_{22}\cdot \sin\theta \cos\theta +\sigma_{33}\cdot \sin\theta \cos\theta +\sigma_{23}\cdot \left(\cos^{2}\theta -\sin^{2}\theta \right)\\ \tau_{n1}\left(\theta \right)=\sigma_{13}\cdot \sin\theta +\sigma_{12}\cdot \cos\theta \end{array},$$
(4)$$\frac{p_{⊥\psi }^{t,c}}{R_{⊥\psi }^{A}}=\frac{p_{⊥⊥}^{t,c}}{R_{⊥⊥}^{A}}\cdot \frac{\tau_{nt}^{2}}{\tau_{nt}^{2}+\tau_{n1}^{2}}+\frac{p_{⊥∥}^{t,c}}{R_{⊥∥}^{A}}\cdot \frac{\tau_{n1}^{2}}{\tau_{nt}^{2}+\tau_{n1}^{2}},$$
and
(5)$$R_{⊥⊥}^{A}=\frac{R_{⊥}^{c}}{2\left(1+p_{⊥⊥}^{c}\right)},$$
where ‘2’ and ‘3’ denote the fiber transverse direction; ‘*θ*’ is the inclined angle of the action plane; the superscript ‘*t*’ denotes tension properties; ‘*c*’ signifies compression properties; $p_{⊥⊥}^{t,c}$ and $p_{⊥∥}^{t,c}$ are material-related inclination parameters; and $R_{⊥}$, $R_{⊥⊥}$, and $R_{⊥∥}$ denote the resistance against $\sigma_{n}$, $\tau_{nt}$, and $\tau_{n1}$, respectively. In this paper, $R_{⊥}^{t}$ and $R_{⊥∥}$ were set as $X_{22}^{t,0}$ (representing the strength corresponding to a deviation of 0.5 MPa from the linear zone) and $S_{12}^{0}$ (indicating the strength at which the secant in-plane shear modulus deviates by 7% from the elastic modulus) to ascertain the initiation of micro-damages [33,39], denoted as point A in Figure 1. Additionally, the values $X_{22}^{t,u}$, $X_{22}^{c,u}$, and $S_{12}^{\;u}$, corresponding to the failure strength of a unidirectional lamina under transverse tension, compression, and in-plane strength, were assigned as $R_{⊥}^{t}$, $R_{⊥}^{c}$, and $R_{⊥∥}$ respectively. These serve to assess the onset of macro-damages, identified as point B in Figure 1.

### 2.2. Damage Evolution

The Cachan CDM model was utilized to depict the evolution of micro-damages in segment AB. The damaged strain energy density $W_{D}$ was rewritten in terms of stresses on Puck’s action plane:(6)$$W_{D}=\frac{1}{2}\left[\frac{\sigma_{11}^{2}}{E_{11}}-\frac{2\upsilon_{12}}{E_{11}}\sigma_{11}\sigma_{n}+\frac{{\langle \sigma_{n}\rangle }_{+}^{2}}{E_{11}\left(1-d_{2}\right)}+\frac{{\langle \sigma_{n}\rangle }_{-}^{2}}{E_{11}}+\frac{\tau_{nt}^{2}+\tau_{n1}^{2}}{2G_{12}\left(1-d_{6}\right)}\right],$$
where $d_{2}$ and $d_{6}$ are micro-damage coefficients reflecting transverse and in-plane shear micro-damages, respectively. A quantity was then defined [21]:(7)$$Y=sup\left(\sqrt{Y_{d^{\prime }}(t)+bY_{d}(t)}\right),$$
where ‘sup’ indicates that *Y* is the maximum value of the loading history; $Y_{d}$ and $Y_{d^{\prime }}$ are the partial derivations of $W_{D}$ with respect to $d_{2}$ and $d_{6}$, respectively:(8)$$\left\{\begin{array}{l} Y_{d}=\frac{\partial W_{D}}{\partial d_{2}}=\frac{1}{2}\frac{{\langle \sigma_{n}\rangle }_{+}^{2}}{E_{11}{\left(1-d_{2}\right)}^{2}}\\ Y_{d^{\prime }}=\frac{\partial W_{D}}{\partial d_{6}}=\frac{1}{2}\frac{\tau_{nt}^{2}+\tau_{n1}^{2}}{G_{12}{\left(1-d_{6}\right)}^{2}} \end{array}\right..$$

The corresponding evolution laws for micro-damages were then written as [21]
(9)$$\left\{\begin{array}{l} d_{2}=\frac{{\langle Y-Y_{0}\rangle }_{+}}{Y_{c}}\\ d_{6}=\frac{{\langle Y-{Y^{\prime }}_{0}\rangle }_{+}}{{Y^{\prime }}_{c}} \end{array}\right..$$

The parameters *b*, $Y_{0}$, $Y_{c}$, $Y_{0}^{\prime }$, and $Y_{c}^{\prime }$ are material-related parameters and can be characterized through cyclic tensile tests of (±45°)_4s_ and (±67.5°)_4s_ symmetric laminates. Details of the procedure for determining the parameters can be found in Ref. [21].

Furthermore, a linear degradation method was employed to illustrate the evolution of macro-damages in segment BC. The degradation coefficients $\eta_{E}$ and $\eta_{G}$, which represent the degradation of the transverse and in-plane shear elastic modulus due to macro-damages, respectively, are expressed as
(10)$$\left[\begin{array}{l} E_{22}\\ G_{12} \end{array}\right]=\left[\begin{array}{l} E_{33}\\ G_{12} \end{array}\right]=\left[\begin{array}{l} \eta_{E}\cdot E_{22}^{0}\\ \eta_{G}\cdot G_{12}^{0} \end{array}\right]=\left[\begin{array}{l} \left(\frac{\left(1-\eta_{rE}\right)\left(f_{E,\mathrm{IFF}}-1\right)}{f_{E,\mathrm{IFF},\max}-1}+\eta_{rE}\right)\cdot E_{22}^{0}\\ \left(\frac{\left(1-\eta_{rG}\right)\left(f_{E,\mathrm{IFF}}-1\right)}{f_{E,\mathrm{IFF},\max}-1}+\eta_{rG}\right)\cdot G_{12}^{0} \end{array}\right].$$

The superscript ‘0’ signifies the tangent modulus at the onset of macro-damages. $\eta_{rE}$ and $\eta_{rG}$ represent the residual stiffness of the transverse and in-plane shear elastic modulus, respectively [40]. Furthermore, $f_{E,IFF,max}$ denotes the maximum allowable stress exposure for IFF.

### 2.3. Plasticity Evolution

As the last component of the constitutive model, an anisotropic elastoplastic model, derived from a one-parameter plastic model proposed by Neto [41], was proposed to capture the plasticity of the matrix. The engineering stress and strain were represented in the vector form:(11)$$\sigma ={\left[\sigma_{11},\sigma_{22},\sigma_{33},\sigma_{12},\sigma_{23},\sigma_{13}\right]}^{T},$$
(12)$$\varepsilon ={\left[\varepsilon_{11},\varepsilon_{22},\varepsilon_{33},\varepsilon_{12},\varepsilon_{23},\varepsilon_{13}\right]}^{T},$$
where the bold font indicates that the variable is in the form of a matrix. To enhance clarity, the five fundamental components of plasticity, i.e., the elastoplastic strain decomposition, an elastic law, a yield criterion stating the yield function, a plastic flow rule defining the evolution of the plastic strain, and a hardening law characterizing the evolution of the yield limit, were systematically presented one after another.

(a)Elastoplastic strain decomposition

The elastoplastic strain **ε** was decomposed into the elastic-strain-coupled damage $\varepsilon^{\mathrm{ed}}$ and the plastic strain $\varepsilon^{p}$. The corresponding rate form is
(13)$$\dot{\varepsilon }={\dot{\varepsilon }}^{ed}+{\dot{\varepsilon }}^{p}.$$

(b)Elastic law

With the assumption of the strain equivalence, the elastic law considering the material damage is
(14)$$\dot{\tilde{\sigma }}=D^{e}\cdot {\dot{\varepsilon }}^{ed},$$
where $D^{e}$ is the Voigt form of the four-order stiffness tensor, and $\tilde{\sigma }$ is the effective stress tensor, given by
(15)$$\tilde{\sigma }=M\cdot \sigma ,$$
where **M** is the damage matrix defined as
(16)$$M=diag\left(\frac{1}{1-d_{1}},\frac{1}{\eta_{E}\left(1-d_{2}\right)},\frac{1}{\eta_{E}\left(1-d_{2}\right)},\frac{1}{\eta_{G}\left(1-d_{6}\right)},\frac{1}{\eta_{G}\left(1-d_{6}\right)},\frac{1}{\eta_{G}\left(1-d_{6}\right)}\right).$$

(c)Yield criterion

The equivalent stress ${\bar{\tilde{\sigma }}}_{y}$ is defined as [42]
(17)$${\bar{\tilde{\sigma }}}_{y}=\sqrt{\frac{3}{2}\left[{\left({\tilde{\sigma }}_{22}\right)}^{2}+{\left({\tilde{\sigma }}_{33}\right)}^{2}+2a{\left({\tilde{\sigma }}_{12}\right)}^{2}+2a{\left({\tilde{\sigma }}_{13}\right)}^{2}+2a{\left({\tilde{\sigma }}_{23}\right)}^{2}\right]},$$
where *a* is a material-related coefficient, reflecting the coupling between the transverse plasticity and in-plane shear plasticity. Then, a quadratic anisotropic yield function Φ was utilized:(18)$$\Phi =\frac{1}{2}{\tilde{\sigma }}^{T}P\tilde{\sigma }-{\bar{\tilde{\sigma }}}_{y}^{2}({\bar{\varepsilon }}^{p}),$$
where **P** is defined as diag(0,3,3,6*a*,6*a*,6*a*). Φ ≤ 0 denotes an elastic state, and Φ > 0 signifies a plastic state. In addition, the relationship between the equivalent stress and the equivalent plastic strain ${\bar{\varepsilon }}^{p}$ was expressed as the exponent form:(19)$${\bar{\tilde{\sigma }}}_{y}({\bar{\varepsilon }}^{p})=\beta {\left({\bar{\varepsilon }}^{p}\right)}^{\alpha },$$
where *α* and *β* are material-related coefficients. The transverse and in-plane shear linear region of FRPs are small and the exponent function can be approximated as linear in that region. Therefore, the initial yield stress was set to zero.

(d)Plastic flow rule

An associative plasticity was presumed, and the evolution of the plastic strain is expressed as
(20)$${\dot{\varepsilon }}^{p}=\dot{\gamma }N,$$
where *γ* is Lagrange’s plastic multiplier; **N** is the flow vector and equals
(21)$$N=\frac{\partial \Phi }{\partial \tilde{\sigma }}=P\cdot \tilde{\sigma }.$$

After definition of the plastic flow rule, the equivalent plastic strain can then be expressed by $\varepsilon^{p}$ under the plastic work equivalence:(22)$${\tilde{\sigma }}^{T}\cdot {\dot{\varepsilon }}^{p}={\bar{\tilde{\sigma }}}_{y}\cdot {\bar{\varepsilon }}^{p}$$

By substituting Equations (18), (20), and (21) into (22), the following was obtained:(23)$${\dot{\bar{\varepsilon }}}^{p}=2\dot{\gamma }{\bar{\tilde{\sigma }}}_{y}=\sqrt{\frac{2}{3}{\left({\dot{\varepsilon }}_{22}^{p}\right)}^{2}+\frac{2}{3}{\left({\dot{\varepsilon }}_{22}^{p}\right)}^{2}+\frac{1}{3a}{\left({\dot{\varepsilon }}_{12}^{p}\right)}^{2}+\frac{1}{3a}{\left({\dot{\varepsilon }}_{23}^{p}\right)}^{2}+\frac{1}{3a}{\left({\dot{\varepsilon }}_{13}^{p}\right)}^{2}}$$

(e)Hardening law

Isotropic hardening was presumed. The internal variable, which in this model was specified as the equivalent strain, is expressed as
(24)$${\dot{\bar{\varepsilon }}}^{p}=\dot{\gamma }H,$$
where *H* is the hardening modulus. From Equations (23) and (24), *H* equals
(25)$$H(\tilde{\sigma })=\sqrt{N^{T}\cdot Z\cdot N},$$
where **Z** is defined as diag(0,2/3,2/3,1/3*a*,1/3*a*,1/3*a*).

## 3. Numerical Implementation Methodology

The material constitutive equations are implemented in Fortran under the Visual Studio framework and linked to ANSYS 19.0 through UserMat, a user-programmable feature of ANSYS. The role of UserMat is to update the Cauchy stress and consistent tangent stiffness by utilizing the received stress, strain, and state variable values of the element at every material integration point during the solution phase. The following sections detail the Cauchy stress solution, consistent tangent stiffness solution, and total numerical algorithm of the developed constitutive model.

### 3.1. Cauchy Stress Solution

The elastoplastic equations present in Section 2.2 are transformed into the backward Euler discretization form:(26)$$\varepsilon_{n+1}^{e}=\varepsilon_{n}^{e}+\Delta \varepsilon -\Delta \gamma Ν\left({\tilde{\sigma }}_{n+1}\right),$$
(27)$${\bar{\varepsilon }}_{n+1}^{p}={\bar{\varepsilon }}_{n}^{p}+\Delta \gamma H\left({\tilde{\sigma }}_{n+1}\right),$$
and
(28)$$\Delta \gamma \geq 0,\;\Phi \left({\tilde{\sigma }}_{n+1},{\bar{\varepsilon }}_{n+1}^{p}\right)\leq 0,\;\Delta \gamma \Phi \left({\tilde{\sigma }}_{n+1},{\bar{\varepsilon }}_{n+1}^{p}\right)=0,$$
where the subscripts ‘*n*’ and ‘*n* + 1’ denote the calculation step of finite element analysis (FEA). To solve this algebraic system, the return mapping algorithm (RMA) is utilized.

At first, it is assumed that the material deformation from step ‘*n*’ to step ‘*n* + 1’ is elastic, and the strain at step ‘*n* + 1’ ought to be
(29)$$\varepsilon_{n+1}^{e,trial}=\varepsilon_{n}^{e}+\Delta \varepsilon$$
and
(30)$${\bar{\varepsilon }}_{n+1}^{p,trial}={\bar{\varepsilon }}_{n}^{p},$$
where the superscript ‘trial’ denotes a trial quantity.

Then, the yield function is calculated with trial strain. If $\Phi^{trial}\leq 0$, no plasticity occurs and the true quantity is the same as that of the trial. Otherwise, plasticity occurs and a ‘plastic correction’ should be executed. The strain at step ‘*n* + 1’ ought to be
(31)$$\varepsilon_{n+1}^{e}=\varepsilon_{n+1}^{e,trial}-\Delta \gamma P\cdot {\tilde{\sigma }}_{n+1}$$
and
(32)$${\bar{\varepsilon }}_{n+1}^{p}={\bar{\varepsilon }}_{n+1}^{p}+\Delta \gamma H\left({\tilde{\sigma }}_{n+1}\right),$$
with
(33)$$\Phi \left({\tilde{\sigma }}_{n+1},{\bar{\varepsilon }}_{n+1}^{p}\right)=0.$$

Multiplying (31) by $D^{e}$, we can obtain the explicit form of the effective stress:(34)$${\tilde{\sigma }}_{n+1}={\left(I+\Delta \gamma D^{e}P\right)}^{-1}\cdot {\tilde{\sigma }}_{n+1}^{trial}.$$

Then, substituting Equations (31), (32), and (34) into (33), a single plastic equation with respect to ∆*γ* of the ‘plastic correction’ is obtained [41]:(35)$$\Phi \left(\Delta \gamma \right)=\frac{1}{2}{\left[{\left(I+\Delta \gamma D^{e}P\right)}^{-1}\cdot {\tilde{\sigma }}_{n+1}^{trial}\right]}^{T}\cdot P\cdot \left[{\left(I+\Delta \gamma D^{e}P\right)}^{-1}\cdot {\tilde{\sigma }}_{n+1}^{trial}\right]-{\bar{\tilde{\sigma }}}_{y}^{2}\left({\bar{\varepsilon }}_{n+1}^{p,trial}+\Delta \gamma H\left({\tilde{\sigma }}_{n+1}\right)\right)=0$$

This equation can be solved by the Newton–Raphson method. The iterative scheme is
(36)$$d\Delta \gamma^{\left(k\right)}=-{\left.\frac{\Phi \left(\Delta \gamma \right)}{\frac{d\Phi }{d\Delta \gamma }}\right|}^{\left(k-1\right)},$$
where ‘(*k*)’ denotes the iterative step, and it is initialized by
(37)$$\Delta {\gamma_{n+1}}^{\left(0\right)}=\Delta \gamma_{n},\;{\bar{\tilde{\sigma }}}_{n+1}^{\left(0\right)}={\bar{\tilde{\sigma }}}_{n+1}^{trial},\;{{\bar{\varepsilon }}_{n+1}^{p}}^{\left(0\right)}={\bar{\varepsilon }}_{n+1}^{p,trial}.$$

The derivative part of Equation (36) is
(38)$$\frac{d\Phi }{d\Delta \gamma }={\left({\tilde{\sigma }}_{n+1}\right)}^{T}\cdot P\cdot \frac{d{\tilde{\sigma }}_{n+1}}{d\Delta \gamma }-2{\bar{\tilde{\sigma }}}_{y}\cdot \alpha \beta {\left({\bar{\varepsilon }}_{n+1}^{p}\right)}^{\alpha -1}\left(H\left({\tilde{\sigma }}_{n+1}\right)+\frac{\Delta \gamma N^{T}}{H\left({\tilde{\sigma }}_{n+1}\right)}\cdot ZP\cdot \frac{d{\tilde{\sigma }}_{n+1}}{d\Delta \gamma }\right),$$
where $d{\tilde{\sigma }}_{n+1}/d\Delta \gamma$ equals
(39)$$\frac{d{\tilde{\sigma }}_{n+1}}{d\Delta \gamma }=-{\left(I+\Delta \gamma D^{e}P\right)}^{-1}\cdot \left(D^{e}P\right)\cdot {\tilde{\sigma }}_{n+1}.$$

Thus far, the Cauchy stress and equivalent plastic strain (used for iterative initialization of the next FEA step) can be updated iteratively until convergence:(40)$${\tilde{\sigma }}_{n+1}^{\left(k\right)}={\left(I+\Delta \gamma^{\left(k\right)}D^{e}P\right)}^{-1}\cdot {\tilde{\sigma }}_{n+1}^{trial},$$
(41)$${{\bar{\varepsilon }}_{n+1}^{p}}^{\left(k\right)}={\bar{\varepsilon }}_{n+1}^{p,trial}+\Delta \gamma^{\left(k\right)}H\left({\tilde{\sigma }}_{n+1}^{\left(k\right)}\right).$$

### 3.2. Consistent Tangent Stiffness Solution

In order to obtain the constituent tangent stiffness, we transfer Equations (26)–(28) into their differential form:(42)$$\left[\begin{array}{ccc} {\left(D^{e}\right)}^{-1}+\Delta \gamma P & 0 & N\\ -\frac{\Delta \gamma \cdot N^{T}\cdot \left(ZP\right)}{H\left({\tilde{\sigma }}_{n+1}\right)} & 1 & -H\left({\tilde{\sigma }}_{n+1}\right)\\ N^{T} & -2{\bar{\tilde{\sigma }}}_{y,n+1}\cdot \alpha \beta {\left({\bar{\varepsilon }}_{n+1}^{p}\right)}^{\alpha -1} & 0 \end{array}\right]\left[\begin{array}{c} d{\tilde{\sigma }}_{n+1}\\ d{\bar{\varepsilon }}_{n+1}^{p}\\ d\Delta \gamma \end{array}\right]=\left[\begin{array}{c} d\varepsilon_{n+1}^{e,trial}\\ 0\\ 0 \end{array}\right].$$

From this, the relationship between the stress and trial strain is derived as
(43)$$\left[{\left(D^{e}\right)}^{-1}+\Delta \gamma P-\frac{\Delta \gamma \cdot \left(NN^{T}\right)\cdot \left(ZP\right)}{H^{2}}+\frac{NN^{T}}{2H{\bar{\tilde{\sigma }}}_{y,n+1}\alpha \beta {\left({\bar{\varepsilon }}_{n+1}^{p}\right)}^{\alpha -1}}\right]\cdot d{\tilde{\sigma }}_{n+1}=d\varepsilon_{n+1}^{e,trial}.$$

Therefore, the consistent tangent stiffness can be deduced according to the chain derivation rule:(44)$$D_{n+1}^{ep}=\frac{\partial \sigma_{n+1}}{\partial {\tilde{\sigma }}_{n+1}}\frac{\partial {\tilde{\sigma }}_{n+1}}{\partial \varepsilon_{n+1}^{e,trial}}=M^{-1}\cdot {\left[{\left(D^{e}\right)}^{-1}+\Delta \gamma P-\frac{\Delta \gamma \cdot \left(NN^{T}\right)\cdot \left(ZP\right)}{H^{2}}+\frac{NN^{T}}{2H{\bar{\tilde{\sigma }}}_{y}\alpha \beta {\left({\bar{\varepsilon }}_{n+1}^{p}\right)}^{\alpha -1}}\right]}^{-1}.$$

### 3.3. Numerical Algorithm

The numerical algorithm for the developed constitutive model is illustrated in Figure 3. The algorithm comprises two key steps: plasticity evolution and damage evolution. The process initiates with the computation of effective stress and elastic strain, which is followed by solving the elastoplastic equations. Subsequently, the damage status is determined based on the 3D Puck criterion, and finally, the damage parameters are updated. In this solving strategy, the damage and elastoplastic variables are subsequently refreshed. The maintenance of satisfactory calculation accuracy could be ensured by judiciously controlling the time step.

## 4. Results and Discussion

### 4.1. Materials Tested and Work Method

To comprehensively validate the constructed constitutive model across various lay-up configurations and stress states, a thorough comparison was conducted between simulations and existing test results of FRP laminates. The assessment focused on three perspectives:
Failure envelope prediction. In this aspect, biaxial tension tests were selected. These tests involved (90°/+30°/−30°)_s_ E-glass/epoxy laminates and quasi-isotropic laminates of (90°/+45°/−45°/0°)_s_ AS4-carbon/epoxy, conducted by Hütter [43] and Swanson [44,45,46], respectively.Deformation prediction. This segment involved the analysis of stress–strain curves of (90°/+45°/−45°/0°)_s_ AS4-carbon/epoxy laminates under a stress ratio of $\sigma_{x}$:$\sigma_{y}$ = 1:20 and 1:2. The *x*-direction was aligned along the direction of the 0° lay-up direction. Christoforou [47] and Trask [48] conducted these tests at the University of Utah, respectively.Open-hole-tension (OHT) strength prediction. In this case, OHT tests of T300-carbon/epoxy laminates conducted by Chang [49,50] were chosen as a reference. These tests encompass three different lay-ups ([0/(±45°)_3_/90°_3_]_s_, [0/(±45°)_2_/90°_5_]_s_, and [0/(±45°)_1_/90°_7_]_s_), each with four different geometric configurations.

The test data from the first two cases were a part of the benchmark data of the WWFE, a famous academic event known as the ’Failure Olympics’ of FRPs. A concise summary of the specimen preparation, test equipment, loading scheme, and final results is available in [7], with comprehensive details provided in the respective original articles. Detailed experimental information of the third case can be found in [49].

The parameters for E-glass/epoxy, AS4-carbon/epoxy, and T300-carbon/epoxy have been compiled in Table 1. Specifically, the elastic modulus, strength, and stress–strain curves of the unidirectional laminae were referenced from [37,50]. Coefficients relevant to damage determination were collated from [5], while damage evolution coefficients were extracted from [21,40]. The plastic evolution coefficient *a* was adopted as recommended in [51]. The current emphasis was directed towards the determination of the values associated with the plastic evolution coefficients, denoted as *α* and *β*. It is widely recognized that unidirectional laminae exhibit significant plasticity in in-plane shear [29,30,31,32,33,34,35]. Drawing upon this recognition, the determination of *α* and *β* can be facilitated through successive adjustments, ensuring a close alignment between the simulated in-plane stress–strain curve and the corresponding experimental observations.

A quarter FEA model was employed for both biaxial tension and OHT conditions, as illustrated in Figure 4a and Figure 4b, respectively. Symmetry boundaries were constrained in their normal direction. SOLID185, a 3D, eight-node, layered solid element in ANSYS, was utilized for meshing. Layer properties, encompassing the layer orientation angle and thickness, were established through section commands (SECTYPE and SECDATA), with three integration points per layer specified. The previously constructed constitutive model was used as the material model.

For the biaxial load condition, the dimensions of the model were set at 100 mm in width and length. The thicknesses of the 90° and 30° E-glass/epoxy layer were 0.172 mm and 0.414 mm [37], respectively, while all layers of the AS4-carbon/epoxy measured 0.1375 mm each [37]. The elements were uniformly meshed, and a mesh sensitivity analysis was conducted, revealing minimal mesh dependency in the present numerical model. No stress concentrations were observed, and the stress distribution within each layer was uniform. A mesh density of 50 × 50 × 1 was deemed sufficient and was consequently employed.

For OHT, all layers of the T300-carbon/epoxy possessed the same thickness, and the overall thickness of the laminate was 2.62 mm [50]. The laminate geometry (i.e., hole diameter, width, and length, respectively) encompassed four categories: 3.18 mm, 19.05 mm, and 177.8 mm; 6.35 mm, 25.4 mm, and 203.2 mm; 3.18 mm, 12.7 mm, and 177.8 mm; and 6.35 mm, 25.4 mm, and 203.2 mm [49,50]. For convenience, these categories are successively labeled A, B, C, and D in the subsequent discussions. Due to the stress concentration near the central hole in the OHT specimen, the meshes within an 8 mm radius of the sample center were refined. Line divisions along the radius and circumference of the quarter circle were set at 20. The longitudinal division of the remaining part of the specimen was established at 30, with a space ratio of 0.6. A mesh sensitivity analysis demonstrated that this mesh density was sufficient. The simulation results were found to be consistent with a model employing twice the mesh density.

### 4.2. Biaxial Tension Failure Envelope

Simulations of biaxial tests were conducted on (90°/+30°/−30°)_s_ E-glass/LY556 laminates. The features of this test are as follows [38]: (1) the biaxial strength under tension–compression loading is lower than the uniaxial strength values, and (2) there is a tendency for an enhancement in the biaxial tensile strength. As is illustrated in Figure 5, the simulation results effectively capture both features, and the projected strength aligns well with the test results.

The biaxial test comparison of AS4/3501-6 (90/+45°/−45°/0°)_s_ quasi-isotropic laminates is depicted in Figure 6. The features of this test are as follows [38]: (1) the biaxial strengths are slightly higher than the uniaxial strength values, and (2) once compression stress in the *x*-direction is applied, the strength in the *y*-direction is reduced below the uniaxial strength. As is shown in Figure 6, the simulation results effectively represent both features, and the projected strength aligns well with the test results, with the exception of slight non-conservation in the second quadrant.

### 4.3. Biaxial Tension Deformation

Simulations of the biaxial tension under a stress ratio of $\sigma_{x}$:$\sigma_{y}$ = 1:20 were conducted on the AS4/3501-6 (90/+45°/−45°/0°)_s_ laminate (Figure 7). This biaxial load nearly equaled the *y*-direction tension. As a result, $\varepsilon_{x}$ exhibited negative results due to the Poisson ratio effect. Figure 7 illustrates that both the simulation curve and experimental test exhibited some nonlinearity. This is attributed to the transverse and shear damages in the 0° and ±45° layers. Simulations also indicated that the 0° layers entered the macro-damage stage at a $\sigma_{y}$ of 256 MPa, and the stress–strain response exhibited a slight deviation from the elastic response (the black dashed line in Figure 7). At 352 MPa, the ±45° layers entered the macro-damage stage, and the deviation was more pronounced. The stiffness attenuation of the 0° layers was notable around 540 MPa (with $d_{2}$, $d_{6}$, $\eta_{E}$, and $\eta_{G}$ of the 0° layers reaching 0.7, 0.8, 0.94, and 0.98 at 544 MPa, respectively). The 90° and ±45° layers would carry a greater load, resulting in a gradual change in the curve slope around this stress level.

The features of this test, as documented in [38], are as follows: (1) the measured stress–strain curves exhibited relatively linear behavior up to final failure, and (2) final failure took place due to fiber tension fracture. In general, the simulations demonstrated good agreement with the test results and their features. The projected strength was determined to be 736 MPa, exhibiting only a marginal deviation of approximately 2.5% from the test strength of 718 MPa. The final failure mode was characterized by fiber tension fracture, and the stress–strain response in the 0–500 MPa range closely matched. The primary discrepancy observed was that the strain beyond 500 MPa was slightly higher than that of the test results. The rationale behind this discrepancy lies in the slower damage evolution of the constructed constitutive model compared to the actual process, resulting in the significant stress redistribution of the simulation occurring slightly later. This discrepancy can be attributed to the inaccuracy of parameters related to damage evolution, which were selected based on values suggested for the general carbon/epoxy material. A more accurate determination of these parameters specific to AS4/3501-6 could enhance the simulation accuracy.

The comparison of biaxial tension under a stress ratio of $\sigma_{x}$:$\sigma_{y}$ = 1:2 for these laminates is illustrated in Figure 8. The simulations revealed that the 0° and ±45° layers entered the macro-damage stage at $\sigma_{y}$ values of 245 MPa and 352 MPa, respectively. Due to the increased tension loads along their fiber direction compared to the previous case, the nonlinearity was not pronounced. However, a noticeable stiffness attenuation was observed in the ±45° layers around 660 MPa (where $d_{2}$, $d_{6}$, $\eta_{E}$, and $\eta_{G}$ of the +45° layers reached 0.55, 0.78, 0.84, and 0.94, respectively). The 90° and 0° layers would bear more load, resulting in a gradual change in the curve slope around this stress level.

The features of this test are as follows [38]: (1) both the longitudinal and transverse strain are positive (tensile); (2) the curves exhibited a slight change in the slope at around 400 MPa of transverse stress; and (3) the transverse curves exhibited a change in the stiffness near the final points. Overall, the simulations aligned well with the test results and their features. The projected strength was determined to be 830 MPa, deviating by only approximately 2% from the test strength of 847 MPa, and the final fracture strain was almost the same. The most notable discrepancy pertained to the strength at which the slope changed. This discrepancy could be attributed to inaccuracies in the parameters related to damage evolution, similar to the previous case. This resulted in the damage evolution of the constructed constitutive model being slower than the actual process.

### 4.4. Open-Hole-Tension Strength

The OHT tensile strength was represented by the boundary–surface tensile load $\sigma_{m}$. The comparison between the simulations and test results are listed in Table 2, alongside the predictions of Chang [50], Tan [52], and Chen [42]. The average absolute prediction error was 5.9%, which is lower than Chang’s 18.2%, Tan’s 18.6%, and Chen’s 8.6%. The precision of the constitutive model has been improved.

Figure 9 depicts the stiffness attenuation of the [0/(±45°)_3_/90°_3_]_s_ laminate labeled B near the open hole. The load direction is in the horizontal direction, and factor ‘1’ indicates that the material is absolutely damaged, while ‘0’ denotes a non-damaged state. Figure 9a illustrates the **D_e_**(1,1) attenuation of the first layer, which is primarily subjected to longitudinal tension. Damage first occurs near the hole at 180 MPa, and as the load increases, the damaged zone diffuses towards the edge of the specimen. Figure 9b,c show the **D_e_**(4,4) attenuation of the second and seventh layers that are mainly subjected to transverse tension and in-plane shear. Damage initially occurs near the hole at approximately 120 MPa, and with an increasing load, the damaged zone diffuses towards the edge until laminate failure occurs. The attenuation of these two laminae is almost the same. Figure 9d shows the **D_e_**(2,2) attenuation of the 10th lamina that is primarily subjected to transverse tension.

## 5. Conclusions

This study presents a novel anisotropic damage–plasticity model for FRPs. All inherent parameters can be readily obtained through basic mechanical tests, making the model practical and user-friendly. Initially, a constitutive modeling method was detailed. The 3D Puck criteria were employed to assess the onset of micro- and macro-damages, initiating the evolution process through a combination of CDM and linear stiffness attenuation methods. The evolution was coupled with a one-parameter plastic model, and five basic components of the plastic model were outlined. Subsequently, the material constitutive equations were implemented in Fortran under the Visual Studio framework and linked to ANSYS 19.0 through UserMat. In a numerical implementation, the Cauchy stress was solved using the RMA, and the consistent tangent stiffness was deduced based on the solved stress and the chain derivation rule. Finally, the developed constitutive model was verified using existing biaxial tension and OHT tests of glass and carbon FRP laminates. The simulations demonstrated a high degree of correspondence with the test results.

Further research will prioritize the precise determination of damage evolution parameters. This study opted for parameters related to damage evolution based on the values suggested by the authors in [21,40]. However, as different FRPs possess specific damage evolution values, inaccuracies in these parameters could lead to deviations, as described in Section 4.3. To address this issue, cyclic tensile tests of (±45°)_4s_ and (±67.5°)_4s_ laminates will be conducted for specific materials (such as AS4/3501-6). Additionally, we also intend to develop a mesoscale model for FRPs to numerically determine these parameters. This could streamline the determination process by obviating the need for cyclic tension tests, requiring only tests on the fiber, matrix, and their interface.

## Figures and Tables

**Figure 1 polymers-16-00334-f001:**
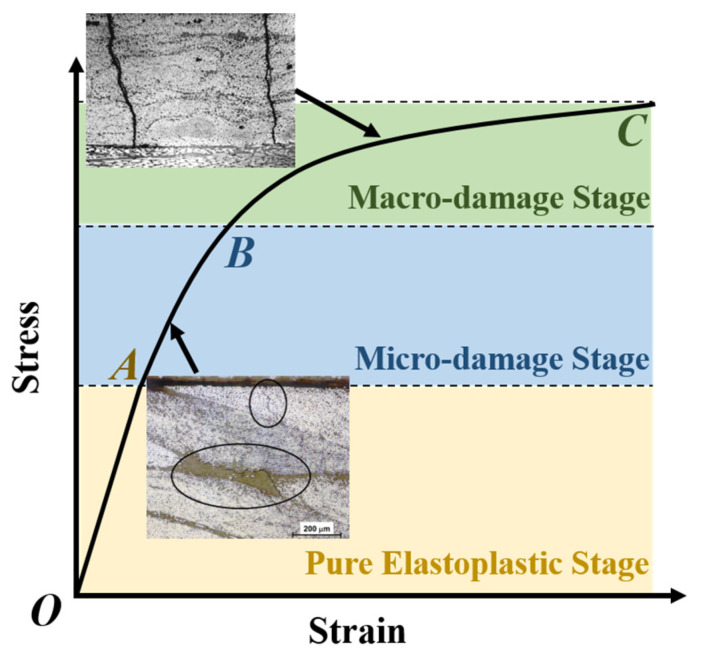
Three stages of the loading process of an FRP unidirectional lamina.

**Figure 2 polymers-16-00334-f002:**
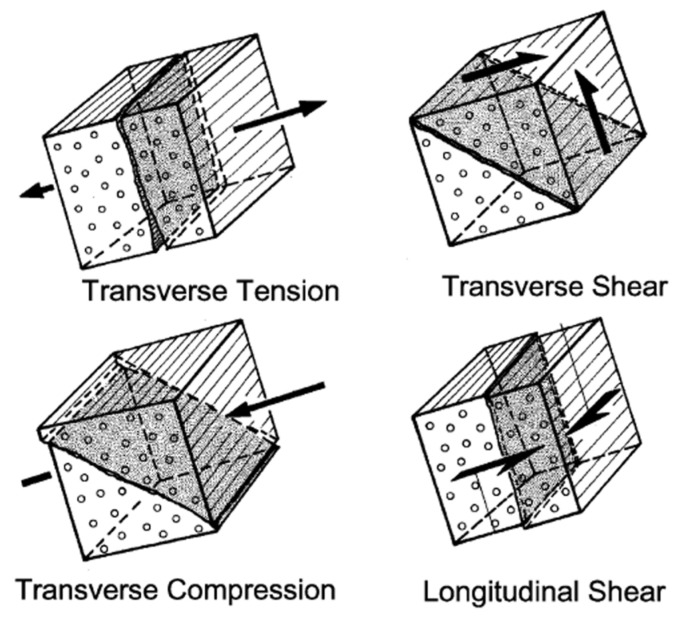
Forms of IFF (Reproduced with permission from [5]).

**Figure 3 polymers-16-00334-f003:**
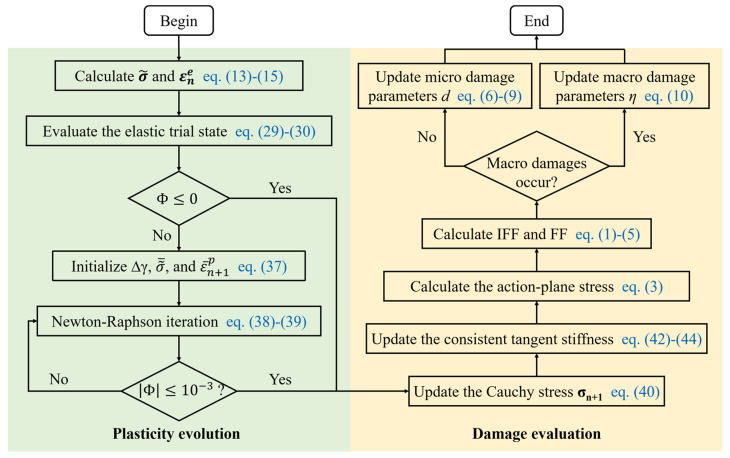
Numerical algorithm of the constitutive model.

**Figure 4 polymers-16-00334-f004:**
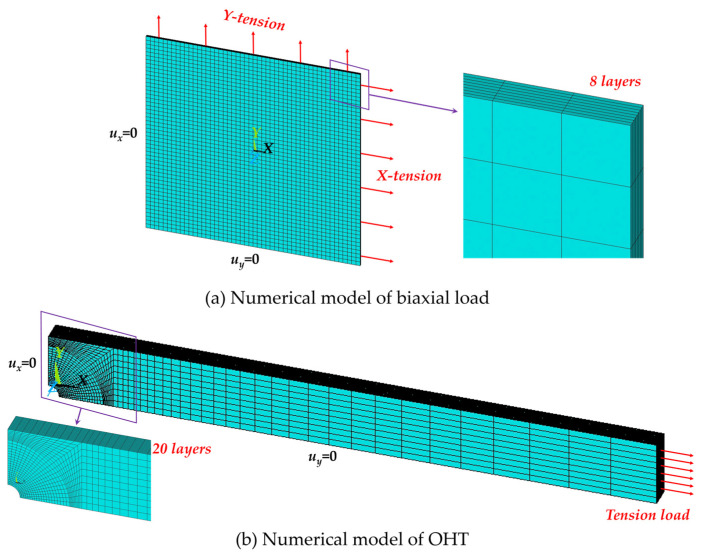
Numerical models of biaxial load and OHT.

**Figure 5 polymers-16-00334-f005:**
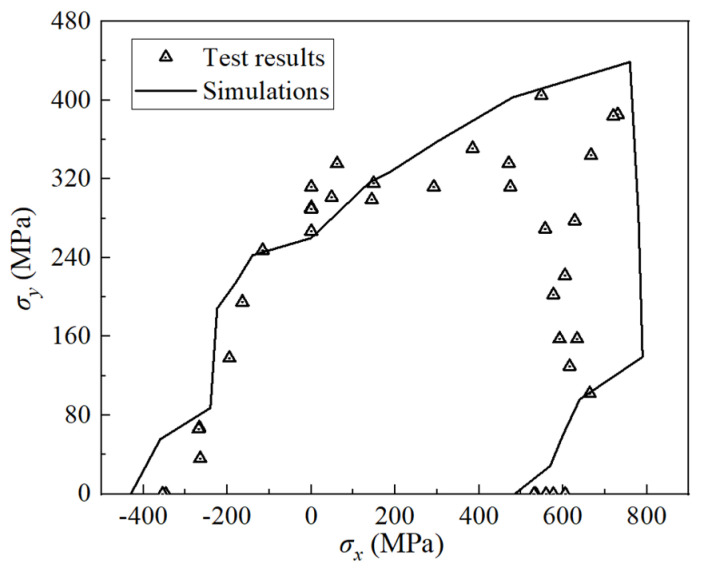
Biaxial tests of (90°/+30°/−30°)_s_ E-glass/LY556 laminates.

**Figure 6 polymers-16-00334-f006:**
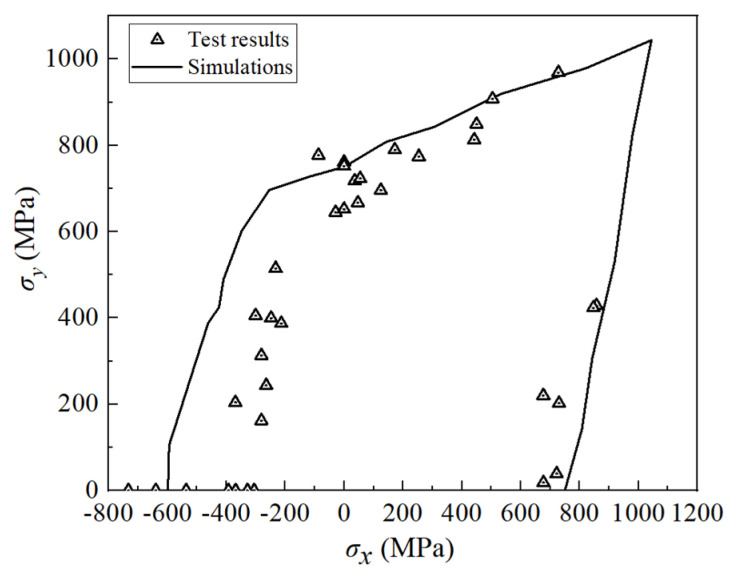
Biaxial tests of (90/+45°/−45°/0°)_s_ AS4/3501-6 laminates.

**Figure 7 polymers-16-00334-f007:**
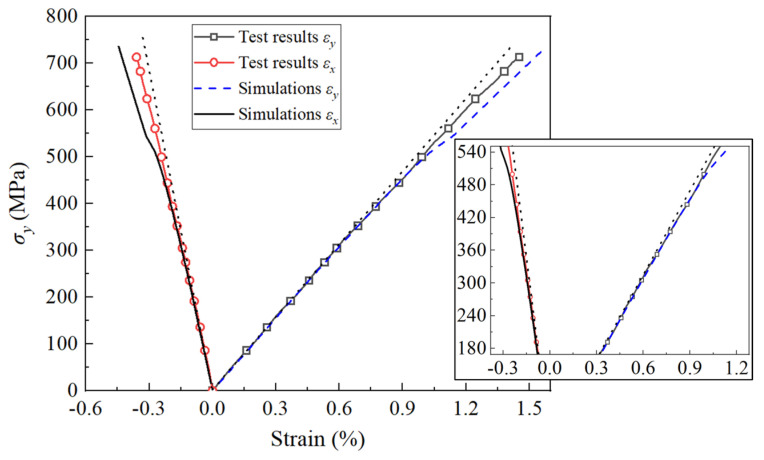
Deformation of (90/+45°/−45°/0°)_s_ AS4/3501-6 laminates under biaxial tension ($\sigma_{x}$:$\sigma_{y}$ = 1:20). The subfigure magnifies the deformation under 180–540 MPa. The black dashed line is the elastic response.

**Figure 8 polymers-16-00334-f008:**
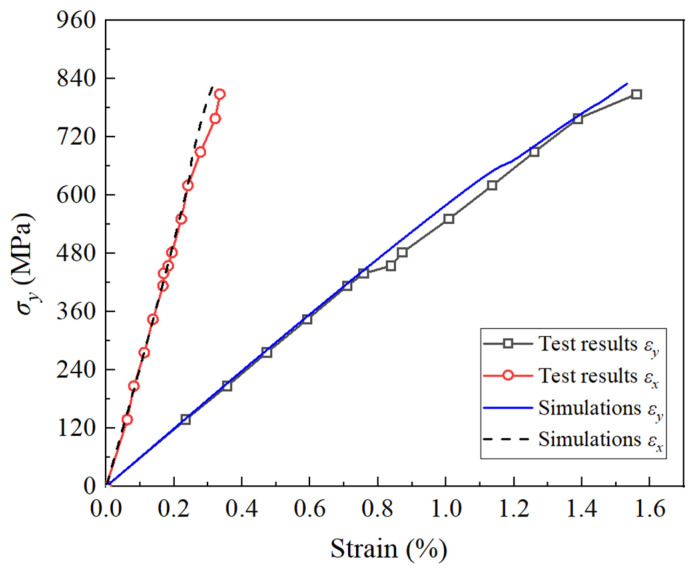
Deformation of (90/+45°/−45°/0°)_s_ AS4/3501-6 laminates under biaxial tension ($\sigma_{x}$:$\sigma_{y}$ = 1:2).

**Figure 9 polymers-16-00334-f009:**
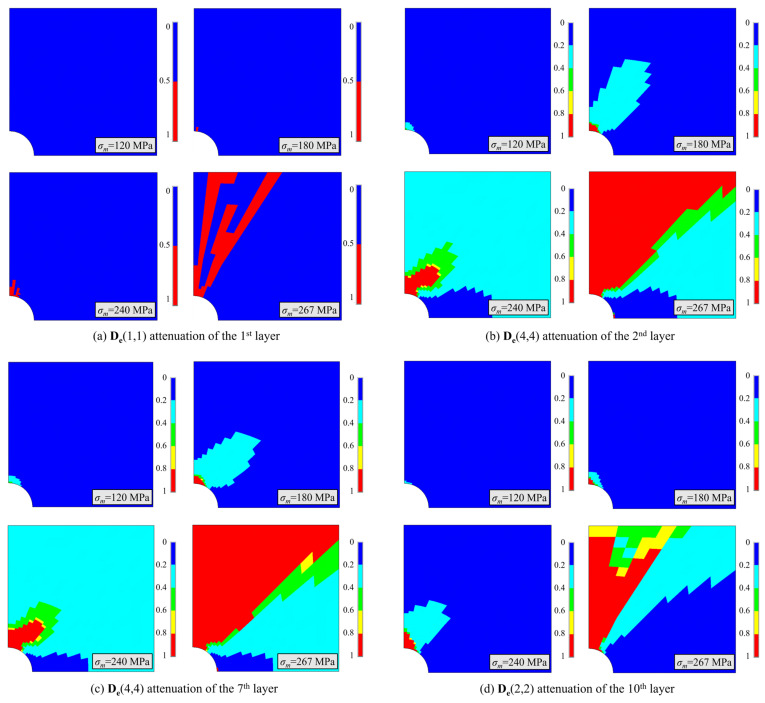
Stiffness attenuation of [0/(±45°)_3_/90°_3_]_s_ laminates labeled B.

**Table 1 polymers-16-00334-t001:** Parameters of three laminae.

Fiber Type	E-Glass 21 × K43 Gevetex	AS4-Carbon	T300-Carbon	Remark
Matrix	LY556/HT907/DY063	3501-6	1034-C	-
Specification	Filament winding	Preprg	Preprg	
Fiber volume fraction	0.62	0.60	0.53	
$E_{11}$ (GPa)	53.48	126	146.86	Elastic characteristics, referenced from [37,50]
$E_{22}$ (GPa)	17.7	11	11.38	
$G_{12}$ (GPa)	5.83	6.6	6.14	
$\nu_{12}$	0.278	0.28	0.30	
$\nu_{23}$	0.4	0.4	0.4	
$X_{11}^{t}$ (MPa)	1140	1950	1730.6	Strength, referenced from [37,50]
$X_{11}^{c}$ (MPa)	570	1480	1379	
$X_{22}^{t,0}$ (MPa)	22.9	31.5	43.8	
$X_{22}^{t,u}$ (MPa)	35	48	66.5	
$X_{22}^{c}$ (MPa)	114	200	268.2	
$S_{⊥∥}^{0}$ (MPa)	37.71	45	56.5	
$S_{⊥∥}^{u}$ (MPa)	72	79	93	
$p_{⊥⊥}^{t}$	0.2	0.3	0.3	Parameters relating to damage determination, referenced from [5]
$p_{⊥⊥}^{c}$	0.25	0.3	0.3	
$p_{⊥∥}^{t}$	0.3	0.35	0.35	
$p_{⊥∥}^{c}$	0.25	0.3	0.3	
*b*	4.4	2.5	2.5	Parameters relating to damage evolution, referenced from [21,40]
$Y_{0}$	0.014	0.24	0.24	
$Y_{c}$	1	3.78	3.78	
$Y_{0}^{\prime }$	0.01	0.15	0.15	
$Y_{c}^{\prime }$	3.24	2.77	2.77	
$f_{E,IFF,max}$	5.0	5.0	5.0	
*a*	2.0	1.25	1.25	Parameters relating to plasticity evolution. *a* was referenced from [51], and *α* and *β* were deduced through in-plane shear tests.
*α*	0.24	0.2	0.08	
*β* (MPa)	1050	1200	3000	

**Table 2 polymers-16-00334-t002:** The strength prediction of T300/1034-C OHT specimens.

Laminate Lay-Up		Tensile Strength $\sigma_{m}$ (MPa)							Error (%)	
		Test	Present	Chang	Tan	Chen	Present	Chang	Tan	Chen
[0/(±45°)_3_/90°_3_]_s_	A	277.2	282.0	227.5	275.8	293.1	1.8	−17.9	−0.5	5.7
	B	256.5	267.6	206.8	275.8	252.2	4.3	−19.4	7.5	−1.7
	C	226.2	236.8	206.8	262.0	269.1	4.7	−8.5	15.9	19.0
	D	235.8	238.0	179.3	248.2	238.3	0.8	−24.0	5.3	1.1
[0/(±45°)_2_/90°_5_]_s_	A	236.5	238.0	193.1	186.2	239.1	−0.6	−18.4	−21.3	1.1
	B	204.1	230.5	172.4	186.2	214.3	13.0	−15.5	−8.8	5.0
	C	177.9	197.2	165.5	172.4	216.3	10.9	−7.0	−3.1	21.6
	D	185.5	188.1	151.7	158.6	205.8	1.4	−18.2	−14.5	11.0
[0/(±45°)_1_/90°_7_]_s_	A	191.0	179.9	144.8	227.5	171.0	−5.8	−24.2	19.1	−10.5
	B	158.6	167.1	124.1	227.5	150.4	5.4	−21.7	43.5	−5.2
	C	134.5	157.5	124.1	213.7	155.0	17.1	−7.7	59.0	15.3
	D	160.0	151.5	103.4	200.0	135.7	−5.3	−35.3	25.0	−15.2

## Data Availability

Data are contained within the article.

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
