# Peer review of "An Anisotropic Damage-Plasticity Constitutive Model of Continuous Fiber-Reinforced Polymers"

_polymers, 2024, doi:10.3390/polym16030334_

Round 1

Reviewer 1 Report

Comments and Suggestions for Authors

The paper is generally well written and the simulation proses and experimental results  are well presented. But there are a few comments that should be corrected to make the paper publishable.

11)      Please, add to the paper how did you determine plastic coefficients listed in table 1.

22)      Add more detailed description of biaxial test procedure. How did you get experimental results?

33)      It should be explained in more details why for theAS4/3501‐6 laminates the εx shows negative strain results.

44)      It should be explained in more details what is the reason of change in  slope of the simulation curves in figs 4 and 5, as well as why the experimental curves in fig 4 don’t show any changes in slope?

55)      There is some illogical structure of the article. You used 3 types of laminates, but section 4.1 is related only on 2 laminates, section 4.2 is related on 1 laminate, section 4.3 describes results for only 1 laminate.

Reviewer 2 Report

Comments and Suggestions for Authors

The article proposes a new mechanical model for the destruction of layered composites based on a polymer binder. The main relationships of the model are presented, a module for the ANSYS finite element complex that implements this model is developed. A comparison with existing experimental data is provided. It shows better agreement with experiment compared to other models.

There are the following comments and suggestions regarding the article:

1. The article contains a large number of formulas and notations, so it is advisable to make a list of notations.

2. The finite element mesh used for the analysis is not shown. It is also not indicated what order of elements were used: linear and quadratic. Has a mesh sensitivity analysis been performed?

3. The conclusions are written in a fairly laconic manner. It is advisable to expand them and indicate prospects for further research.

Reviewer 3 Report

Comments and Suggestions for Authors

Dear authors,

The topic of the research of the manuscript is interesting for theoretical modelling of the continuous fiber-reinforced polymers which has an elasto-plastic behaviour.

The authors must clarify some details and data, which are missing.

In my opinion, the entire text of the manuscript must be re-written and re-organized and then, it can be resubmitted again for review.

In order to improve this manuscript, I recommend a lot of major changes and improvement as shown below.

1.The purpose and the main objectives of this research must be highlighted better in Abstract and in the end of Introduction section.

2. The authors must introduce a list of notations and abbreviations for all quantities which are used in equations because otherwise it is difficult for the readers. There are many symbols and notations which are not described in the main text of the manuscript.

3. The letter W is usually used for strain energy.

4. The authors should explain what does it mean inner fiber fracture (IFF) and they can use a drawing if it is required.

5. Please, explain how the Eq. (2) was obtained.

6. What do the notations d2 and d6 represent in Eq. (9).

7. Please, revise and re-organize the theoretical model given in sections 2 and 3 because it is difficult to understand and to follow for the readers.

8. The authors should introduce a new section Materials tested and work method” in order to describe all materials tested and the equipment used in mechanical tests. The authors made a mixture between presentation of the tests and results in section 4 without presenting the materials tested, the equipment and testing parameters. A scheme of loading in mechanical tests is also required. The authors must also show what properties are recorded.

9. Description of the analysed materials, given in Table 1, is too general. More details are required: type of fiber reinforcement, type o resin, content of fibers, number of layers, orientation for fibers in each layer and so on.

10. How the coefficients given in Table 1 were computed?

11. The authors must also describe the numerical model used and the improvement added by this research, in the sub-section Work method.

Comments on the Quality of English Language

Dear authors,

Please revise carefully the text of your manuscript and improve the expression in English. 

Round 2

Reviewer 3 Report

Comments and Suggestions for Authors

Dear authors,

I read carefully the revised version of your manuscript and your responses for the comments of the reviewers. The manuscript was improved indeed according to the suggestions and recommendations of the reviewers.

Although the authors have made major improvements and changes, there are still some uncertainties, which are presented below, and some minor changes are required for this reason.

1. The list of notations and abbreviations for all quantities must be presented in alphabetical order.

2. According to the author’s response Eq. (2) is based on Ref. [5]. Please, cite Ref. [5] in the text before Eq. (2). Please, revise all equations of the manuscript and cite all references required.

3. In Table, the text “Elastic modulus” must be replaced with “Elastic characteristics” because this refers to both elastic moduli and Poisson’s ratios (transversal contraction coefficients).

4. The authors must cite the references used for the material properties, in caption of Table 1.

Comments on the Quality of English Language

Dear authors,

Please revise carefully the text of your manuscript and improve the expression in English. 
